

**Tracer experiment and model evidence for macrofaunal shaping of microbial**
**nitrogen functions along rocky shores**
Catherine A. Pfister[1], Mark A. Altabet[2], Santhiska Pather[2], Greg Dwyer[1]
[1] Department of Ecology and Evolution, University of Chicago, Chicago IL, USA
[2] School for Marine Sciences and Technology, University of Massachusetts, Dartmouth,
Bedford, MA, USA
Correspondence to: C. A. Pfister (cpfister@uchicago.edu)



**Abstract.** The interdependence of macrofauna and microbes is increasingly recognized as a key
feature affecting nutrient cycling. Tidepools are ideal natural mesocosms to test macrofauna and
microbe interactions, and we quantified rates of microbial nitrogen processing using tracer
enrichment of ammonium ($^{15}N_{NH4}$) or nitrate ($^{15}N_{NO3}$) when tidepools were isolated from the
ocean during low intertidal periods. Experiments were conducted during both day and night as
well as in control tidepools and those from which mussels had been removed allowing us to
determine the role of both animal presence and daylight in microbial nitrogen processing. We
paired time-series observations of $^{15}N$ enrichment in $NH_4^+$, $NO_2^-$, and $NO_3^-$ with a differential
equation model to quantify multiple, simultaneous nitrogen transformations. Mussel presence
and daylight increased remineralization and photosynthetic nitrogen uptake. When we compared
ammonium gain or loss that was attributed to microbes versus photosynthetic uptake, microbes
accounted for 32% of this ammonium flux on average. Microbial transformations averaged 61%
of total nitrate use; thus, microbial activity was almost 3 times photosynthetic nitrate uptake.
Because it accounted for processes that diluted our tracer, our differential equation model
assigned higher rates of nitrogen processing compared to prior source-product models. Our in
situ experiments showed that animals alone elevate microbial nitrogen transformations two
orders of magnitude, suggesting that coastal macrobiota are key players in complex microbial
nitrogen transformations.

Keywords: tide pools, enrichment experiment, *Mytilus californianus*, differential equation
model, nitrification, nutrient fluxes




## 1. Introduction

Nitrogen cycle processes are carried out by a diversity of taxa, from microbes to macrofauna, that can all reside in the same habitat. Nevertheless, most studies tend to focus on characterizing and/or measuring the rate of only a single transformation at a time (e.g. nitrification or nitrate reduction), despite the co-occurrence of a diversity of nitrogen processes including those leading to loss or retention. Given an anthropogenic doubling over the past century of the supply rate of biologically available nitrogen to ecosystems (Galloway et al. 2008, Fowler et al. 2013) simultaneous with accelerated harvest of animals that recycle nitrogen (Worm et al. 2006, Maranger et al. 2008), it is essential that we understand the interacting contributions of microbes and macrobiota to nitrogen cycling. Using the experimental tractability of rocky shore tidepools as natural mesocosms, coupled with isotope tracer enrichments and mathematical modeling, we estimate here the rate of simultaneous nitrogen transformations as a function of animal abundance and time of day.

Along upwelling shores, the paradigm of productivity driven by upwelled nitrate has been challenged by studies quantifying the effects of animal excretion and regeneration (Dugdale and Goering 1967, Aquilino et al. 2009, Pather et al. 2014). It is well known that nitrogen regeneration is quantitatively significant in a variety of ecosystems (Schindler et al. 2001, Vanni 2002, Layman et al. 2010, Subalusky et al. 2014), However, to make a significant contribution to productivity, uptake of animal excreted ammonium by photo- and chemolithotrophs needs to be sufficiently rapid to retain nitrogen locally to avoid dispersion into the larger environment.



Microbial nitrogen transformations are diverse, converting inorganic nitrogen among different
biologically available ($NH_4^+$, $NO_2^-$, or $NO_3^-$) or unavailable ($N_2$) forms.  Accordingly, the
relative importance of these pathways also influences the retention or loss of regenerated
nitrogen. In coastal environments, there is increasing documentation that microbial nitrogen
transformation (e.g. chemolithotrophs) is intimately associated with nitrogen-regenerating
animals (Welsh and Castadelli 2004, Pfister et al. 2010, Heisterkamp et al. 2013, Stief 2013). We
know little about how these associations may impact nitrogen uptake by autotrophs. Rapid use of
animal-regenerated ammonium is likely by both obligate ammonia oxidizing microbes (e.g.
Ward 2008) as well as phototrophs that prefer it for energetic reasons (Magalhães et al. 2003,
Zehr and Kudela 2011). Accordingly, ammonium production by animals may be an important
contributor to the productivity along rocky shores of the northeast Pacific that are part of the
California Current Large Marine Ecosystem (CCLME).

There is parallel evidence that marine animals host diverse microbiomes (Pfister et al. 2010,
2014b, Miranda et al. 2013, Moulton et al. in press) as well as stimulate phototrophs with
excreted nitrogen (Taylor and Rees 1998, Plaganyi and Branch 2000, Bracken 2004). However,
we know little about the magnitude of animal impacts on nitrogen transformations. Incubating
seawater or sediment separate from the natural environment has provided controlled estimates of
single nitrogen transformations (e.g. Yool et al. 2007). However, a principle challenge has been
quantifying in situ the simultaneous nitrogen transformations that characterize natural
communities. Animal species may host nitrogen-metabolizing microbes while phototrophs in the
same environmental setting simultaneously compete for the animals' excreted ammonium. Light
levels controlling phototroph ammonium uptake may thus mediate nitrogen transformations.




Here we quantify the influence of a common coastal marine animal, the California mussel, on the
overall magnitude of and the partitioning between simultaneous nitrogen transformations, using
tidepools at low tide as 'experimental mesocosms'. We use an experimental approach to test the
possible interacting roles of this animal and light on the rates of nitrogen transformations that, in
particular, influence net nitrogen retention. We manipulated the presence and absence of mussels
and light in combination with stable isotope tracer addition to directly test their effects on
nitrogen transformations. As with some other tracer experiments, (Tank et al. 2008, Peterson et
al. 1997), there were multiple fates for a single tracer addition, necessitating the use of
differential equations to simultaneously quantify multiple, simultaneous nitrogen processes.  By
fitting model parameters with experimental data, we derive estimates for microbial nitrogen
transformations that are much higher than published rates where rate estimates are treated
singularly. We use the experiment and model to test whether nitrogen transformations in the
tidepools are elevated by mussels, inhibited by light, or affected by other environmental
variables. We also test for evidence of interactions between phototrophs and nitrogen-utilizing
microbes.

**2. Methods**
**2.1 A Model for experimental data**
Stable isotope enrichment experiments are an established methodology for quantifying nitrogen
processing in marine environments where the transfer of a tracer between source and product
pools is measured over time (Glibert et al. 1982, Lipschultz 2008).  Typically, these assays are
done on seawater or sediments (e.g. review by Beman et al. 2011), though there are some



examples where an organism is assayed (e.g. Heisterkamp et al. 2013). One acknowledged
challenge of these experiments is the simultaneous occurrence of multiple processes that can
dilute isotopically the source pool. For example, in a $^{15}NH_4$ tracer to estimate nitrification, the
ammonium tracer could be diluted by the production of unlabeled $NH_4^+$ by remineralization or
the microbial reduction of nitrite.  Without accounting for isotope dilution, rates of transfer of
$NH_4^+$ to other pools would accordingly be underestimated.  To assess the importance of isotope
dilution in our tidepool systems we compare rates of nitrogen transformation using two
approaches: (1) Using the previously used source-product model for a single transformation from
a $^{15}N$-enriched source to a single product which does not account for isotope dilution (as
discussed in Glibert et al. 1982, Lipschultz 2008), and (2) Using a set of 6 differential equations
for modeling six different, simultaneous nitrogen processes which accounts for isotope dilution
in all relevant pools as well as the passage of tracer into intermediate pools.

The fates of 3 forms of inorganic nitrogen (ammonium, nitrite, nitrate) in an isolated tidepool
include a variety of processes mediated by microbes and other intertidal inhabitants, and are
illustrated in Fig 1.  For ammonium, increases in concentration (and dilution of an enriched
tracer) can occur via excretion by animals and is represented by remineralization ($m$).
Phototrophs, both prokaryotic and eukaryotic, can assimilate ammonium and nitrate leading to
decreases in concentration, designated by uptake ($u$). Microbial transformations include
ammonium oxidation to nitrite ($h$), nitrite oxidation to nitrate ($x$), nitrate reduction to nitrite ($y$),
and nitrite reduction to ammonium ($r$). The parameter, $m$, is represented as a constant rate in our
model system whereas the other parameters are first-order rate constants in which the rate is the
product of the constant and the appropriate concentration. Because they are the first steps toward




denitrification (production of $N_2$), both nitrite and nitrate reduction should be favored under low
oxygen conditions. Denitrification, in its entirety, nor anammox (which combines ammonium
and nitrite to produce $N_2$), are not explicitly modeled. Experiments to date that have utilized gas-
tight chambers have not detected nitrogen loss via $N_2$ gas production (unpublished data) and we
thus assume that nitrate and nitrite reduction were incomplete with respect to $N_2$ production and
consistent with nitrogen retention in the system.

The traditional source-product model generally involves estimating an average rate from time 0
to time t (Lipschultz 2008) and has the general form:
$Rate = (R_k(t) - R_k(0))/[(R_s(0) - R_k(0)) * \Delta t] * [\bar{k}]$                Eq. (1)
where $k$ is the sink or product at time $t$ (or the average $\bar{k}$), $s$ is the source. Average product
concentration over the source of the experiment is $\bar{k}$ and $R$ designates the atom % ($^{15}N/(^{15}N$
$+^{14}N$) x100) of either the source or product component at the beginning of the experiment (0) or
the end (t).  Equation (1) can be used to estimate individual nitrogen transformation rates
assuming little change in $R_k$. For example, ammonium oxidation to nitrite (*h* in Fig 1) is
estimated by adding $^{15}NH_4$ and monitoring the $^{15}N$ enrichment in nitrite. Nitrate reduction to
nitrite (*y*) is estimated by adding $^{15}NO_3$, and monitoring the $^{15}N$ enrichment in nitrite.

A recognized shortcoming of Eq. (1) is that multiple simultaneous processes (e.g. Fig. 1) can
change the concentration and isotopic composition of source and product nitrogen pools
(Lipschultz 2008). Resolving the influence of multiple, contemporaneous nitrogen
transformations requires a new approach that accounts for their influence over time on the
distribution of $^{15}N$ tracer. Pather et al. (2014) used an isotope dilution model (Glibert et al. 1982)



that included simultaneous ammonium remineralization and uptake. Here, we extend that
approach by constructing a differential equation model that includes all six simultaneous
processes described above. We then fit our model to observed time-dependent changes in the
concentrations and isotopic composition of ammonium, nitrite and nitrate. Because microbial
metabolisms ($h, x, y, r$), phototroph uptake ($u$), and animal metabolism ($m$) should be occurring
simultaneously, a major advantage of the differential equation model is that it estimates multiple,
simultaneous processes.

In our differential equation model (Fig. 1), three differential equations describe how the
concentrations of ammonium (A), nitrite (Ni), and nitrate (Na) in nmol L$^{-1}$ change with time as a
function of the 6 nitrogen flux terms.

$$\frac{dA}{dt} = m + r(Ni) - h(A) - 2u(A) \qquad \text{(Eq. 2)}$$

$$\frac{dNi}{dt} = h(A) + y(Na) - r(Ni) - x\,(Ni) \qquad \text{(Eq. 3)}$$

$$\frac{dNa}{dt} = x(Ni) - y(Na) - u(Na) \qquad \text{(Eq. 4)}$$

Ammonium remineralization ($m$) is assumed to be a constant rate independent of ammonium
concentration. However, the other fluxes are first-order dependent on source concentrations with
$h$, $u$, $r$, $x$, and $y$ as the rate constants for ammonium oxidation, phototroph uptake, and nitrite
reduction, nitrite oxidation, and nitrate reduction respectively. We also assumed that ammonium
uptake ($2u$) was double that of nitrate uptake, a ratio reflecting the relative energetic ease of
ammonium uptake by phototrophs (Thomas and Harrison 1985, Dortch 1990) and supported by
measurements (Hurd et al. 2014). This 2:1 multiplier fit the data well across tidepool
experiments and provided better fits than a higher or lower multiplier for ammonium:nitrate



uptake. We note, however, that there are likely among species differences in *u* and its multiplier
for ammonium uptake that need further study in marine macroalgae. By using only *u* to represent
both phototrophic ammonium and nitrate uptake, we avoided an increase in the number of
parameters and we simplified our model fitting routine. Although we initially set *u* to zero at
night, we found that model fits were best when we let the model fit some phototrophic uptake at
night, a phenomenon consistent with the observation that dark photosynthesis via carbon storage
occurs in intertidal macroalgae (Kremer 1981). We excluded the uptake term (*u*) from nitrite
dynamics because nitrite is at much lower relative abundance compared with ammonium and
nitrate and is not known as a preferred nitrogen source for phototrophs. Finally, we note that *u*
could also include uptake by heterotrophic bacteria. Based on the results presented below,
however, phototrophic uptake appeared to dominate the *u* term. Given that nitrate and nitrite
reduction are favored only at low $O_2$ concentration, it might be presumed that reducing processes
are insignificant. However, tidepools with their natural complement of animals and algae,
sediment, and small nooks and crannies likely have a high degree of spatial heterogeneity in
oxygen and our results show significant rates of these processes.

Three equations model the time-varying concentrations (nmol $^{15}$N L$^{-1}$) of $^{15}$N ammonium
(n15A), nitrite (n15Ni), and nitrate (n15Na). $^{15}$NH$_4$ is diluted over time by remineralization (*m*)
in the naturally occurring ratio of $^{15}$NH$_4$ to $^{14}$NH$_4$ of 0.00366. All other fluxes transfer $^{15}$N from
source to product in proportion to total nitrogen transfer:

$$\frac{dn15A}{dt} = m(0.00366) + r(n15Ni) - h(n15A) - 2u(n15A) \qquad (Eq.\,5)$$

$$\frac{dn15Ni}{dt} = h(n15A) + y(n15Na) - r(n15Ni) - x(n15Ni) \qquad (Eq.\,6)$$



$$\frac{dn15Na}{dt} = x(n15Ni) - y(n15Na) - u(n15Na) \qquad (Eq.\,7)$$

All parameter definitions are summarized in Table 1. Although isotope fractionation is known to
occur for these nitrogen transformations, their magnitude is small compared to experimental
enrichment values (e.g. Granger et al. 2008, Casciotti 2009, Granger et al. 2010, Swart et al
2014). We thus assumed that fractionation was insignificant in the context of this experimental
manipulation and that first order reaction rate coefficients were equivalent for $^{14}$N and $^{15}$N
containing forms of DIN.

We solved Eqs. 2-7 for the 6 parameters (*m, u, h, x, r, y*) simultaneously, by finding the best fits
to the concentration and $^{15}$N data for each experimental tidepool (see Sect 2.3). We further
leveraged this experimental approach by comparing results for experiments carried out during
the day and at night, and in tidepools with and without mussels, generating multiple parameter
estimates and analyzing how they varied with environmental variables. To do so, we conducted
all 4 experimental variants in each tidepool over the course of 2 months (daytime $^{15}$NH$_4$,
nighttime $^{15}$NH$_4$, daytime $^{15}$NO$_3$, nighttime $^{15}$NO$_3$) (see Methods below).

**2.2 Isotope enrichment experiments in tidepools**
All isotope enrichment experiments were done in tidepools at Second Beach, a rocky north-
facing bench 2 km east of Neah Bay, WA, USA (48°23' N, 124° 40' W) within the Makah Tribal
Reservation. The experimental methods were described in Pather et al (2014), but are briefly
reviewed here.  Since 2002, California mussels (*Mytilus californianus*) have been removed from
5 tidepools while 5 others have remained as controls; in the year of this study, mussels were
hand-removed (by cutting byssal threads) a month prior to the experiment to eliminate any



biogeochemical signal of our presence. Besides this single perturbation, the pools have been left
intact and contain a natural assemblage of macroalgae, microphytobenthos, surfgrasses
*Phyllospadix scouleri* and *P. serrulatus* and macrofauna such as limpets, anemones, and fishes;
the tidepools were 1.2 to 1.5 m above Mean Lower Low Water (MLLW) (Pfister 2007).  The
isolation of these tidepools for 5 to 6 hours during the low tide excursions both during daylight
and nighttime hours during the summer of 2010 made it ideal to use the tidepools as intact
mesocosms and probe the nitrogen transformations in natural ecosystems.

Four $^{15}$N enrichment experiments within these 2 groups of tidepools provided a test of the fate of
ammonium and nitrate, as a function of day and night hours (e.g. with and without
photosynthesis), and the presence and absence of animals.  The '$\delta$' notation is standard for
expressing relatively low levels of $^{15}$N enrichment as well as variations in natural abundance $^{15}$N
($\delta^{15}$N‰=[(R$_{sample}$-R$_{atmN2}$)/R$_{atmN2}$] x 1000) and is used here for expressing measured values. For
model calculations, $\delta^{15}$N values were first converted to $^{15}$N/$^{14}$N ratios and then to the
concentration of $^{15}$N by multiplying by the corresponding nutrient concentration. The four
enrichment experiments included a target 1000‰ enrichment of either $\delta^{15}$NH$_4$ (added as 0.05M
ammonium chloride, $^{15}$NH$_4$Cl) or $\delta^{15}$NO$_3$ (added as 0.05M sodium nitrate, Na$^{15}$NO$_3$), thus
doubling either the $^{15}$N-NH$_4^+$ or $^{15}$N-NO$_3^-$ concentration during both a daytime low tide (25 Jun
2010, ~0715 to 1245 and 27 Jun 2010, ~0730 to 1300h), and a nighttime low tide (~2000 to
0145h on 13-14 Aug 2010 and 2150 to 0400h on 15-16 Aug 2010). A six-week interval between
daytime and nighttime experiments was necessary due to the timing of low tides in the region.
Strong nighttime low tide excursions only occurred in August, while daytime spring tides are
ideal in June.  These two experimental timepoints showed similar starting tidepool seawater





temperatures (11.4 in Jun versus 11.3$^{o}$C in Aug) and similar DIN concentrations (20.0 and 23.1
$\mu$molL$^{-1}$). Both ammonium and nitrate concentrations in tidepools are typically high (>10
$\mu$molL$^{-1}$) minimizing any concentration-related effects from tracer addition. Tidepool volume
had been estimated previously with addition of a known amount of blue food coloring (e.g.
Pfister 1995) and averaged 57.1 L with a range of 26.1 to 97.4 L. Deviations in our target of
1000‰ initial enrichment occurred due to natural variation in nutrient concentrations at the time
of tracer addition, as well as error in tidepool volume estimates.

In all experiments, a water sample prior to tracer addition was collected to verify natural
abundance isotope levels (T$_o$). After tracer solution was added and stirred, a sample of water was
immediately taken to estimate actual initial enrichment (T$_1$). A second sample was taken ~ 2 h
later (T$_2$), followed by a final sample after ~5 h (T$_3$), resulting in 3 samples to estimate the time
course of concentration and $^{15}$N enrichment in NH$_4^+$, NO$_2^-$, and NO$_3^-$ in tidepool water. Although
it would have been ideal to have greater than 4 samples to precisely describe the time course of
$^{15}$N through time, this number represented a cost-effective number across ten replicate tidepools
and four experiments, and minimized investigator disturbance during the experiment. For each
sample, we filtered ~180 ml of tidepool water through a syringe-filter (Whatman GF/F) into
HDPE bottles, which we kept frozen until analysis.  All nutrient concentrations were analyzed at
the University of Washington Marine Chemistry lab, while isotope determinations were done at
University of Massachusetts, Dartmouth. Methodology for nutrient and isotopic composition was
reported previously (Pather et al. 2014, Pfister et al. 2014a).  Nighttime sampling was done using
headlamps, and took only 2-5 min, resulting in negligible illumination near tidepools. Tidepool



oxygen, pH and temperature (Hach HQ4D) were also collected at ~ 2 h intervals throughout the
experiment, and all tidepools had a HOBO temperature logger recording at 10 min intervals.

**2.3 Fitting the Differential Equation Model to Data**
Each tidepool experiment had 3 time points for nitrogen isotope composition and concentration,
allowing parameter fits to be made to our model for each experiment. We solved our differential
equations using the 'ode' function of R (in the deSolve R package, Version 3.1.0, www.r-
project.org, Soetaert et al. 2012). The fit of our model to the data was calculated with the
'modCost' function of the FME package, which calculates the sum of the squared errors between
the model and the data. We fit the model to the data using the 'modFit' function that uses a
Levenberg-Marquardt minimization algorithm (Soetaert et al. 2010). As we did this estimation
for each experiment, not treatment averages, we were able to examine stoichiometric
relationships between nitrogen fluxes maintained at the scale of individual tidepools. Though the
fitting routine always converged, we further tested the robustness of the fitting routine in several
ways. First, we randomly varied the initial values for the parameters 100 times, drawing initial
values from uniform distributions that allowed the parameter estimates to vary over several
orders of magnitude (between 0 to 10). Because the $m$ parameter was not first order and logically
could be large, it ranged from 0 through $10^6$. In all cases, the sum of squares of at least the best
10 parameter sets were within $10^{-3}$ (or less than 1-3% different), strongly suggesting that our
fitting routine found the best parameter sets. As a second test of the model, we calculated net
production or loss of $^{15}N$ by comparing the resulting total moles of $^{15}N$ from the observed values
in each tidepool at the end of each experiment to the corresponding best-fit parameter estimates.



Finally, we compared our differential equation model with the source-product model shown in
Eq. (1). Because our tracer experiments had 3 time points ($T_1$, $T_2$, $T_3$), we used the interval from
$T_1$ to $T_2$ to estimate the first paths for the transfer of tracer via oxidation or reduction (*h* and *y*)
and the interval from $T_2$ to $T_3$ to estimate the second oxidation or reduction process (*x* and *r*). In
this way, there was time for the tracer to become incorporated into nitrite before we estimated
the transformation rates of nitrite oxidation (*x*) in the case of enriched ammonium addition, or
reduction (*r*) in the case of enriched nitrate addition. Focusing our source-sink estimation on
these intervals allowed us to detect the greatest rate estimates from the source-sink model.

We measured multiple responses in our experimental manipulation. We analyzed all responses
with a linear mixed effects model using tidepool as a random effect and testing for a statistical
interaction between mussel presence and light (R, www.r-project.org).

**3. Results**
**3.1 Isotope Patterns in experiments**
After approximately 5 to 6 hours of isolation at low tide, results were dependent on both the
presence of mussels and the availability of sunlight (Fig. 2, Table 2). Ammonium concentration
was overall greater with mussels and during the day, and oxygen, temperature and pH all tended
to be greater during the day. Tidepool pH was lower at night ($p<0.05$) and possibly lower with
mussels ($0.10<p<0.05$). The dynamics of $\delta^{15}N_{NH4}$, $\delta^{15}N_{NO2}$, and $\delta^{15}N_{NO3}$ over the course of the
experiment revealed transfer of the tracer isotope and thus the action of microbial nitrogen
transformations. When $^{15}N\text{-}NH_4^+$ was added, enrichment in $\delta^{15}N_{NO2}$ and $\delta^{15}N_{NO3}$ was seen,





though the presence of mussels diluted the $\delta^{15}N_{NH4}$ signal. Similarly, enrichment in $\delta^{15}N_{NH4}$ and
$\delta^{15}N_{NO2}$ followed the addition of $^{15}N\text{-}NO_3^-$ (Fig S1).

**3.2 The differential equation model estimates nitrogen transformation rates**
The advantage of using our tidepool experiments is that they contain the full range of actual
biological components and environmental fluctuations; but as they vary in the composition of
these components they also show individual differences. We thus fit the model to each tidepool
individually, rather than a mean value, allowing any influences due to environmental differences
to be incorporated into parameter estimates. ODE model predictions were generally highly
concordant with the observed nutrient and isotope data measured for each tidepool experiment
(Fig. 3). In addition to providing a good visual fit to the data for each tidepool (Fig. 3), the
estimated parameters predicted well the total amount of $^{15}N$ measured at the end of the
experiment (Fig. 4). Individual results deviated by as much as $\pm20$ nmol L$^{-1}$, but the estimated
and measured quantities were very similar and indicated the model showed no bias toward
producing or consuming $^{15}N$ (Fig. 3).  The mean $^{15}N$ was 122.3 nmol total in the ammonium
enrichment experiments and 158.6 total in the nitrate enrichment experiments, indicating that
deviations were relatively modest (<16%), especially given the multiple sources of variability in
collecting and analyzing tidepool seawater.

**3.3 The significant effect of mussels and light on nitrogen processing**
The rates of ammonium remineralization in tidepools that we estimated with our ODE model
were greatest during the day when mussels were present, as was the uptake of ammonium (Fig.
2). In turn, all nitrogen metabolisms showed the greatest rates in the presence of mussels (Fig. 5,



Table 3, Table 4).  Further, all nitrogen transformations were greatest during the day with the
exception of nitrate reduction. For ammonium and nitrite oxidation ($h$A and $x$Ni), rates increased
an order of magnitude in the presence of mussels and during the day.  As with all the microbial
transformations, nitrogen uptake attributed to all photosynthesizing species, from microalgae to
macroalgae and seagrasses, was greatest with mussels and also during the day. When we tallied
the percentage of ammonium flux due to microbes (nitrification + nitrite reduction) relative to all
the ammonium flux per tidepool (Table 3), we found that microbial ammonium flux accounted
for 32% of all ammonium flux when mussels were present and it was daylight. Similarly,
microbial nitrate flux was 61.4% of all nitrate flux. Although inorganic nitrogen concentrations
were always greater with mussels (Fig. 2), the rates of nitrogen transformations we estimated
were greatly affected by time of day and mussels (Figs. 2, 5, statistical summary in Table 4).

**3.4 Comparing the ODE model to single rate, source-sink models**
All rates of nitrogen transformation during the day and with mussels estimated with our
differential equation model (Eqs. 2-7) were greater than estimated by the traditional source-
product model (Fig. 5, Table 4).  The ODE model always produced an estimate of the
ammonium oxidation rate far greater than that of the source-product model, particularly during
the day. The ammonium oxidation rate estimated with our differential equation model was
uncorrelated with the estimates from the source-product model (Spearman's r=0.004, Table 4).
Overall, there was little concordance between microbial nitrogen transformations estimated with
the ODE model and the source-product model, as the ODE model frequently estimated higher
rates (Fig. 5, Table 3).



**3.5 Inferences about the relationships among nitrogen processes**

Parameter estimates from our model allowed us to assess the potential interaction among

nitrogen processes. We tested how model estimates of photosynthetic versus microbial

chemolithotrophic nitrogen use were related. If competition for ammonium occurs, then

ammonium oxidation ($h$) could be negatively related to phototrophic ammonium uptake ($2u$). To

avoid correlating parameters estimated simultaneously from the same model fitting attempt, we

correlated ammonium oxidation ($h$A) from the $^{15}NH_4$ enrichment with the uptake ($u$) from the

$^{15}NO_3$ experiments (and vice versa) and did not find a significant relationship in either case

(r=0.320, p=0.169 and r=0.297, p=0.200). The significant and positive relationship between

ammonium oxidation ($h$A) and remineralization ($m$) estimated from our differential equation

model (0.656, p<0.001) is likely not an artifact of unidentified parameters in the model. As

evidence, we note that ammonium oxidation in our ODE model was also positively related to

animal remineralization estimated independently, using the simple isotope dilution model from

Pather et al. (2014) (r=0.687, p<0.001). The positive relationship was unaffected by day or night,

indicating no enhancement of ammonium oxidation when photosynthetic ammonium uptake was

minimized.

Finally, we found few correlations between nitrogen transformation rates and oxygen,

temperature and pH in tidepools at the end of the low tide period. Only remineralization and

nitrogen uptake rates show a positive correlation with higher temperatures (r=0.423, p=0.009 &

r=0.432, p=0.008, respectively), primarily eukaryotic metabolic processes that increased with

temperature.



381

## 4. Discussion

### 4.1 Animal and microbial contributions to nitrogen transformations

The remineralization of ammonium, oxidation and reduction of inorganic nitrogen, and the

uptake of ammonium and nitrate were all greater in tidepools with mussels versus those where

mussels were removed. Mean nitrate flux due to microbial processing (the sum of microbial

nitrate transformations in Table 3) ranged from 8 to 61% of the total nitrate uptake attributed to

both microbes and phototrophs, with the highest values when mussels were present and it was

daylight.  Microbial processing accounted for an average 32% of the total ammonium flux with

mussels and daylight. Processing of both nitrate and ammonium by microbial chemolithotrophs

was thus significant in this rocky shore environment, and especially so when mussels were

present. Previous analysis of ammonium uptake in this system indicated that suspended particles

(e.g. phytoplankton) in tidepool seawater account for a negligible amount of ammonium uptake

(only 1-3 nmol $L^{-1}$ $h^{-1}$) and microbial activity in tidepool seawater was an order of magnitude

less than benthic microbial activity (Pather et al. 2014). Additionally, benthic algae uptake rates

(estimated at ~5 x $10^{-4}$ $h^{-1}$, Pather et al. 2014) likely dominate the parameter $u$, though the

biomass specific uptake rates for the algae in our tidepools are unknown because we would have

had to destructively sample all algae to estimate this. However, published rates of ammonium

uptake in red algae ranged from 15900-62000 nmol per hour for every gram of algal dry weight,

while those for nitrate are 9700-28500 (Hurd et al. 2014). Thus, several individual algae could

account for the uptake of nitrogen that is not microbial, and our estimates of uptake using the $u$

parameter in the model are consistent with literature values (Table 3).  In total, our enrichment

experiments indicate that microbial transformations can be as great as and even exceed the





contributions of phototrophs to nitrogen dynamics. Further, the microbial activity related to
nitrogen cycling is primarily in association with benthic animals and phototrophs.

Previous genomic analyses showed that inert substrates (e.g. rocks) in tidepools with mussels
host a nearly identical microbial community to those in tidepools without mussels (Pfister et al.
2014b), while mussel shells themselves host a rich diversity of nitrogen-metabolizing microbial
taxa (Pfister et al. 2010). Combined with the nitrogen processing rates we quantified here, these
studies suggest that California mussels are loci for the microbial processing of nitrogen. Marine
invertebrates as hosts for significant nitrogen processing is further supported by work with snails
and other bivalves, which are demonstrated sites of nitrogen transformations including
ammonium oxidation (Welsh and Castadelli 2004, Stief et al. 2009, Heisterkamp et al. 2013).
$N_2O$ production is also suggested for sediment-dwelling bivalves (Heisterkamp et al. 2013) and
those in sealed chambers (Stief et al. 2009). Evidence for bivalves as hotspots for nutrient
dynamics also includes species in river and stream environments (Atkinson & Vaughn 2014).
Mussels on rocky shores can average very high densities of 4661 individuals per $m^2$ (Suchanek
1979), suggesting that ammonium concentrations above mussels should be in mmol
concentrations (Pfister et al. 2010). Observation of much lower concentrations directly over
mussel beds (Aquilino et al. 2009), and in tidepools (this study) suggests the simultaneous
operation of other N processing pathways as observed here.

**4.2 Microbes contribute to nitrogen retention**
In high-energy coastal environments, animal regenerated ammonium could be advected by
waves and currents rather than retained. Because the rates we quantified are rapid, and because

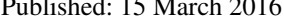



tidepool habitats are high flow refugia, net retention of inorganic nitrogen in nearshore areas can
result, a phenomenon that is likely to enhance local primary production. Over a diel cycle, both
ammonium and nitrite oxidation and nitrate and nitrite reduction occurred, and all are processes
that retain dissolved and biologically available nitrogen. Although we did not follow our tracer
into all tidepool species, previous analyses showed it was readily incorporated into tidepool algae
(Pather et al. 2014).  Nitrogen loss processes were not quantified, though other experiments with
gas-tight chambers indicated no loss of nitrogen via enriched $N_2$ gas (Pfister & Altabet
unpublished data). Additionally, if the loss of $^{15}N$ signal was occurring due to anammox or
denitrification completed to nitrogen gas, then our models would have systematically estimated a
loss of $^{15}N$, a result not supported by our analyses (Fig. 4). Further, phototrophic uptake of $^{15}N$
was the only other term in the model for nitrogen loss. Our model predictions for uptake no only
were robust in both day and night experiments (Fig. 4), but the uptake rates were highly
consistent with measured uptake rates of marine algae (see section 4.1 above). We recognize,
however, that nitrogen loss processes via the production of the greenhouse gas nitrous oxide is
suggested in association with other animal species (Heisterkamp et al. 2013).  Though the return
of nitrogen gas to the atmosphere is a significant feature of low oxygen, open ocean areas (Ward
2013) there was no evidence for it here. In this study, and in the analysis of naturally occurring
nitrogen isotopes (Pfister et al. 2014), nitrogen retention is instead suggested in high-energy
coastal areas, though the generality of this finding deserves further study.

Both nitrate and nitrite reduction rates were significant and are evidence for incomplete
denitrification or DNRA processes thought to be occurring only at low oxygen. Even during
daytime periods of high oxygen, nitrate and nitrite reduction were observed, suggesting that



tidepools provide microsites where these microbial reducing processes can take place. The
oxidation of ammonium and nitrite, though not positively related to final oxygen level, was
greatest during the day and with mussels. Even at night when oxygen could be very low, there
was sufficient ambient oxygen to permit nitrification. Thus, even though remineralization
decreased at night and oxygen levels dropped, ammonium oxidation remained at an average of
160.6 nmol $L^{-1}$ $h^{-1}$ in the presence of mussels.

Although competition for ammonium between nitrifiers and phototrophs is poorly understood,
the preference for ammonium uptake may make it a contested resource.  Sediment microalgae
have been shown to be competitively superior to ammonium oxidizing bacteria, likely due to
higher specific uptake rates and faster growth (Risgaard-Petersen et al. 2004). Here, we found
little evidence for competitive interactions for either ammonium or nitrate between
photosynthetic processes and microbial chemolithotrophs.  Microbial transformations in the dark
did not increase, suggesting that microbial nitrogen metabolism is driven more by the stimulation
of animal excretion that occurs in these tidepools during the day, perhaps because of increased
tidepool temperature (Bayne & Scullard 1977). We also show no evidence of UV inhibition of
nitrification (e.g. Horrigan & Springer 1990, Guerrero & Jones 1995).  We note also that
tidepool ammonium levels rarely were lower than several μM, and thus ammonium should not
have been depleted and limiting unless there are depleted microsites. Further studies at low
ammonium, including areas where animal regeneration is reduced and ammonium may be
contested, are warranted to understand how phototrophs and chemolithotrophs interact.






**4.3 The Differential equation model captures rapid and simultaneous processes**
We developed the ODE model to simultaneously estimate multiple microbial transformation
rates and thus provide a more realistic descriptor of microbial activity in nature. Our model focus
on the rates of simultaneous nitrogen transformations assures that it is general and applicable to
any system. A key result here is that rate estimates from the differential equation model were
often much greater than those from the source-product model (Lipschultz 2008, and Glibert
1982). We suggest two reasons that our ODE estimated greater rates. First, the rapidity of
microbial transformations combined with the diversity of microsites in nature mean that tracer
enrichment can readily cycle through multiple products. Thus, $^{15}$N in ammonium may be
oxidized not only to nitrite, but also to nitrate and then potentially reduced (Fig. 1). Our model
allows this 'cycling', whereas a source-sink model assumes a single source and product are
involved in the estimation of $^{15}$N dynamics. The second reason our ODE model estimates greater
rates than a source-sink is that ammonium remineralization by macrobiota in nature can rapidly
dilute the $^{15}$NH$_4^+$ signal. A diluted $^{15}$NH$_4^+$ signal leads to underestimation of nitrogen dynamics
with source-sink models, a concern noted by its authors when source-product models were
derived. Here, and in Pather et al. (2014), we note that the effects of ammonium dilution were
most pronounced with mussels during the day, where all microbial rates were underestimated
with a source-product model. Our ODE model, in contrast, accounts for the propagation of tracer
dilution by ammonium remineralization to all DIN pools, likely resulting in greater estimates for
multiple nitrogen metabolisms. Indeed, our estimates of nitrification are several orders of
magnitude greater than those estimated in other coastal locales with source-product models
(Beman et al. 2011), allowing us to conclude that macrobiota greatly enhance rates of nitrogen
transformations. We further note that the rates we quantified are characterized by high variability



among tidepools, a result likely due to some measurement error for $^{15}$N enriched field samples,
but also from natural variability in space and time for processes sensitive to species composition
and environmental factors.

**5. Conclusions**
Tidepools demonstrated a range of prokaryotic and eukaryotic nitrogen metabolisms that varied
with animal presence and the time of day, echoing other recent studies that demonstrate marine
animals serve as sites for a diversity of nitrogen metabolisms (Fiore et al. 2010, Heisterkamp et
al. 2013). The ubiquity in the coastal environment of the flora and fauna found in tidepools
suggests that microbial nitrogen transformations are not unique to tidepools but a general feature
associated with macrobiota. The relatively high variability in the estimates of all microbial
nitrogen transformations we documented is paralleled by variability in the environmental
variables (e.g. oxygen, pH, temperature, species composition) that may also foster a rich mosaic
of tidepool microsites for microbial biogeochemical processing and nitrogen regeneration and
retention.  Scaling up to the entire rocky shore ecosystem suggests a large potential role for
animals in ameliorating fluctuations in upwelling and nutrient delivery. Meanwhile,  ongoing
animal harvest in ocean systems has greatly impacted nitrogen cycling (e.g. Maranger et al.
2008), making it imperative to understand the links between nitrogen in coastal systems and
animal harvest.

*Author Contribution.* CAP, MA, SP designed the experiments and CAP, MA, SP carried them
out and did laboratory analyses. CAP, GD, MA developed the model. CAP prepared the
manuscript with contributions from all co-authors.






*Acknowledgements.* We thank E. Altabet, S. Betcher, B. Colson, M. Kanichy, O. Moulton, P.
Zaykowski for making the field experiment a success, including R. Belanger's lab efforts. K.
Krogsland provided nutrient analyses and J. Larkum laboratory isotope expertise. We thank the
Makah Nation for access. Funding was provided by NSF-OCE 09-28232 (CAP), NSF-OCE 09-
28152 (MAA), and a Fulbright Foreign Student Award (SP).

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




**Figure Captions**
Fig.1. A schematic of the nitrogen cycling model used in this study, where microbial processes
include $h$ as ammonium oxidation, $y$ as nitrate reduction, $r$ and $x$ are nitrite reduction and
oxidation, and $u$ is uptake by phototrophs. These parameters are all first order rate coefficients
and instantaneous fluxes are the product of the parameter and its substrate concentration.
Remineralization, $m$, is a fixed rate. All parameters are defined in Table 1.

Fig. 2. The ending measured concentrations (in µM) for ammonium, nitrite, and nitrate and the
ending seawater temperature ($^o$C), percent oxygen, and pH in all experimental tidepools used for
the Linear Mixed Effect model results in Table 2. The right 3 panels are rates (nmol L$^{-1}$ h$^{-1}$)
estimated from the ODE model (Eqs. 2-7), including the estimated rate of remineralization ($m$)
and ammonium and nitrate uptake rates in experimental tidepools. The dark horizontal line is the
median, the box encompasses 50% of the data and the unfilled circles are outliers. The positive
effect of mussels (shaded bars) on these 3 rates was greatest during the day. Linear mixed effects
model results are in Table 4.

Fig. 3. ODE modeled $^{15}$N fits to the data for 6 representative tidepools in all four enrichment
experiments. The ODE model was fit individually to each tidepool, designated with unique
colors and symbols. Measurements are shown with symbols, while model fits at each time point
are designated with lines; filled symbols with solid lines are 3 separate tidepools with mussels,
while open symbols with dashed lines are 3 tidepools where mussels were removed. The lines
thus represent the differential equation model (Eqs. 2-7) fit based on the modCost function using
sum of squares. The symbols are the measured values (in nmol $^{15}$N L$^{-1}$) for the corresponding



tidepool at each time point; note difference in axes for nitrite. Note that although tidepools
differed greatly in their nutrient dynamics, the model fits are generally close to the measured
value.

Fig. 4. The relationship between the predicted total $^{15}$N (in nmol L$^{-1}$) (by the ODE model) and
observed quantity of total $^{15}$N (in nmol L$^{-1}$) at the end of each of the $^{15}$NH$_4$ and $^{15}$NO$_3$ tracer
experiments. The 1:1 line is shown and indicates that the model did not, on average, lead to an
artificial production or loss of $^{15}$N and thus provided a reasonable fit to overall $^{15}$N dynamics.
Each estimate is per tidepool and filled symbols are night, while unfilled symbols are day.

Fig. 5. The estimated rates (nmol L$^{-1}$ h$^{-1}$) of microbial nitrogen transformations based on the
ODE model in the left panel (Eqs. 2-7) and the source-product model (Eq. 1; e.g. Lipschultz
2008) on the right. A. ammonium oxidation ($h\bar{A}$), B. nitrite oxidation ($x\overline{Ni}$), c. nitrate reduction
($y\overline{Na}$), d. nitrite reduction ($r\,\overline{Ni}$). Note differences in axes; the differential equation model rates
are shown at 4 times the scale of the source-sink model. All other legend elements as in Fig. 2.








Fig. 1

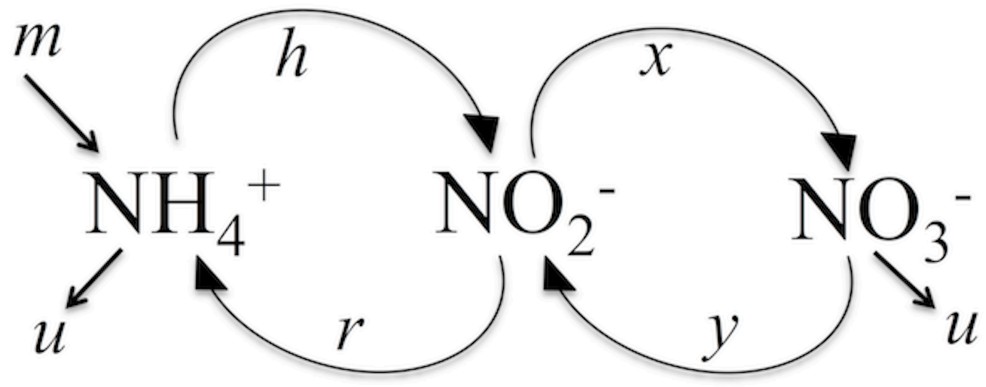







Fig. 2

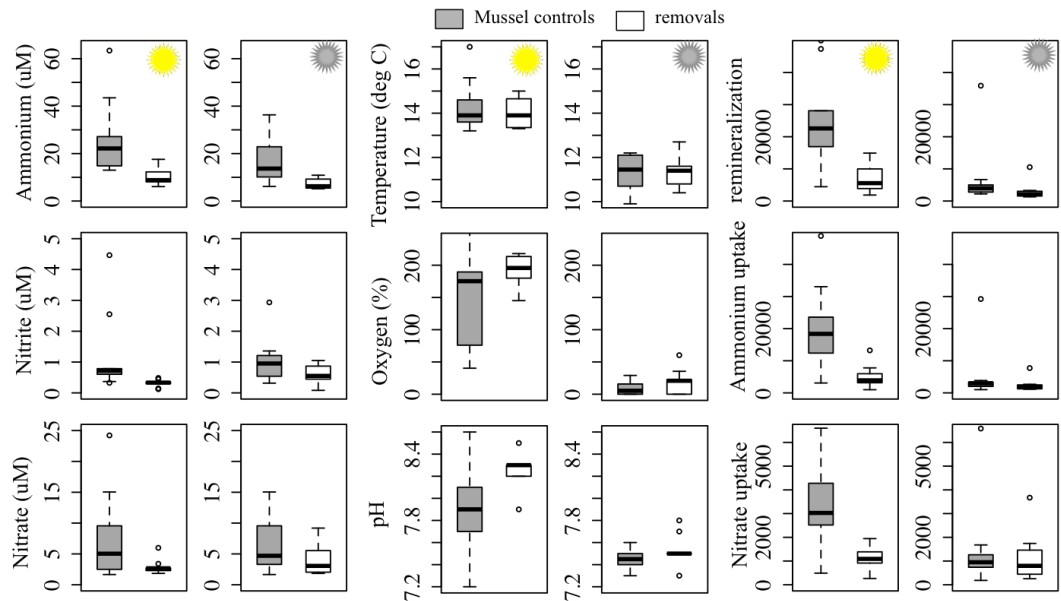
















Fig. 3

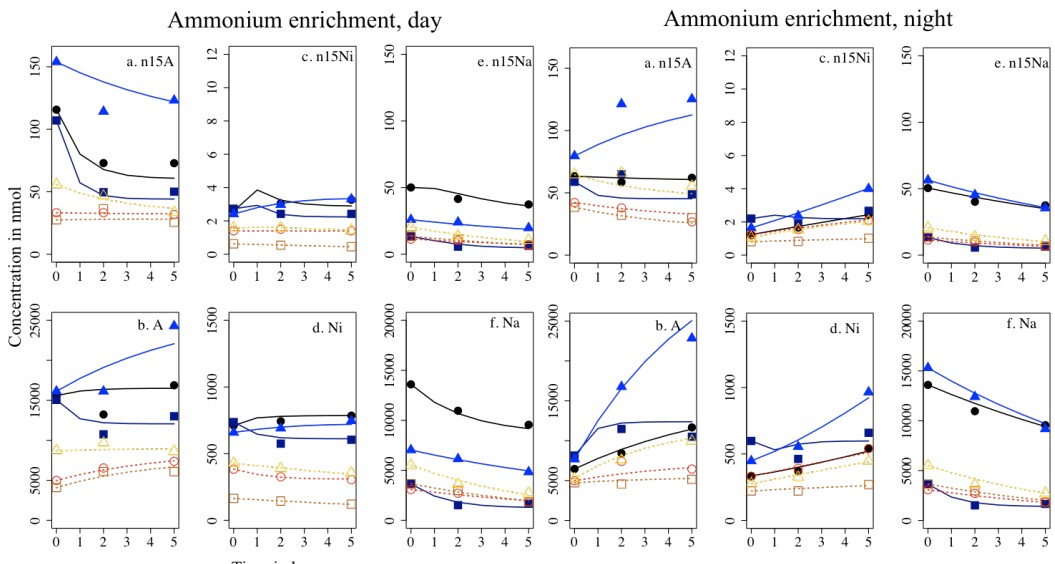





Fig.4

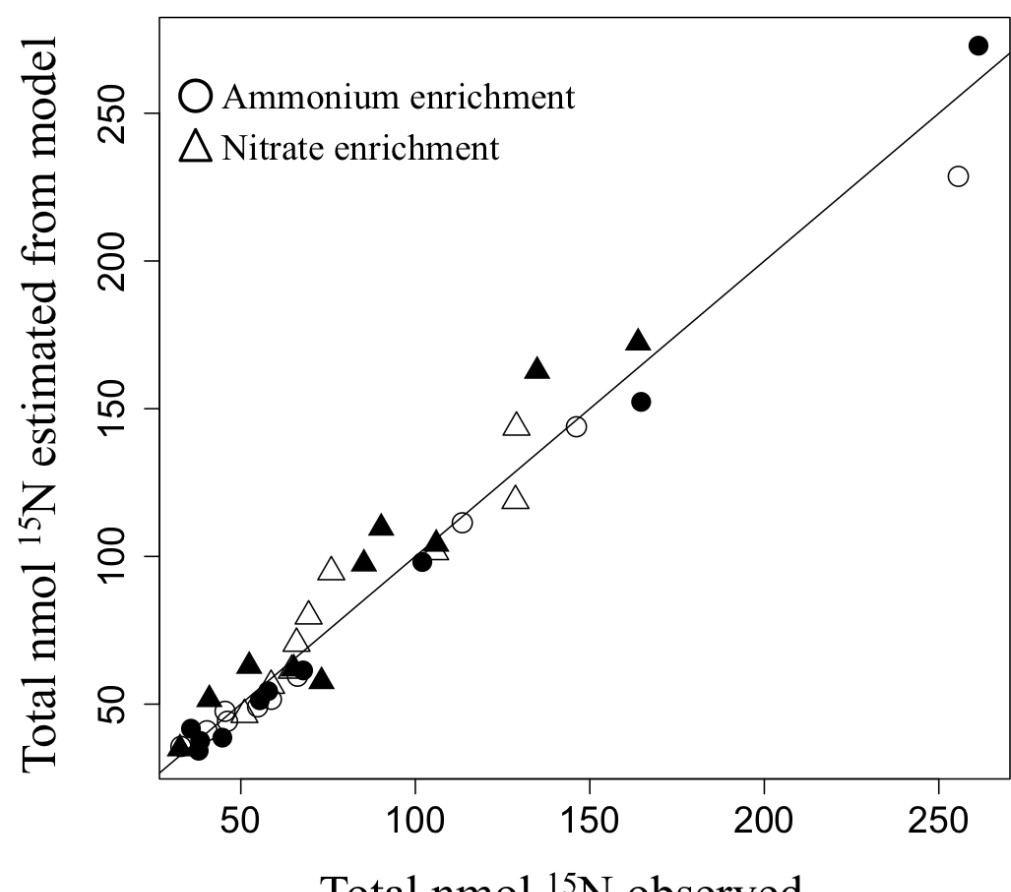





Fig. 5

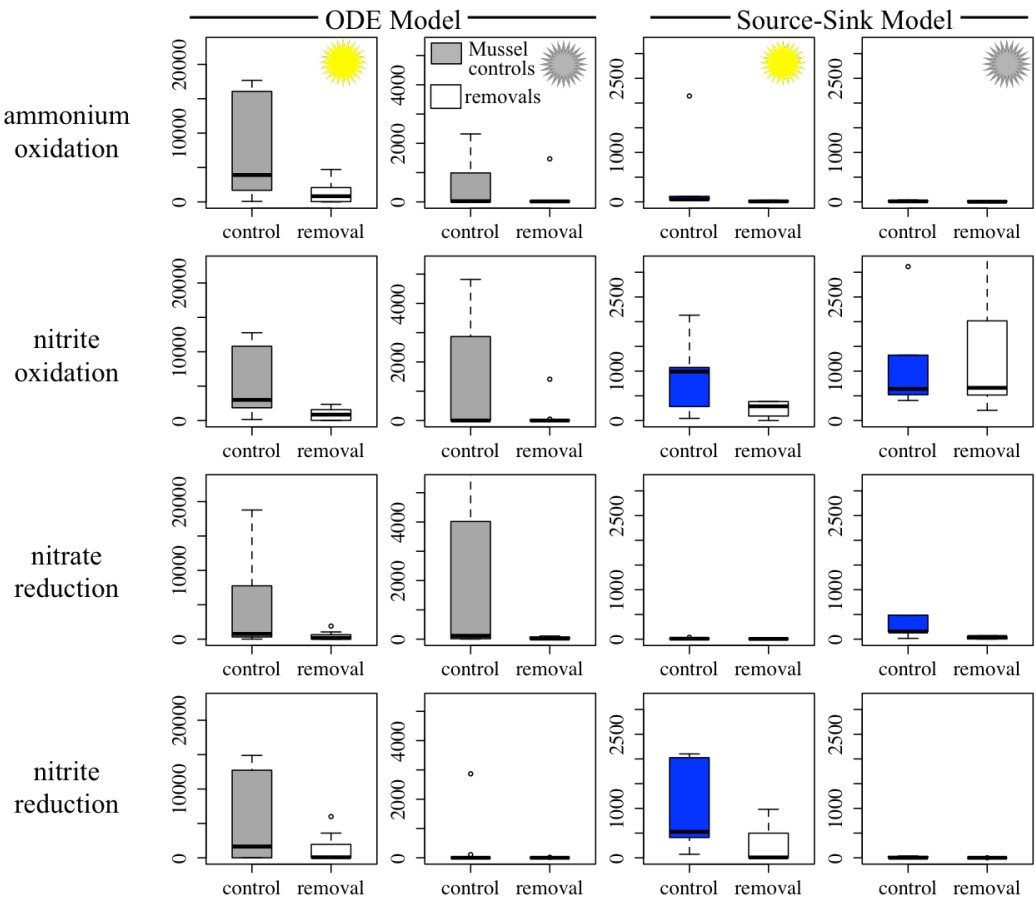



Table 1. A list of observed and modeled parameters used in this study.

| Parameter | Definition | Method of estimation |
|---|---|---|
| $\delta^{15}N‰$ | $[(R_{sample} - R_{atmN2})/R_{atmN2}] \times 1000$, where R is $^{15}N/^{14}N$ | Direct experimental measurement |
| A | ammonium concentration (nmol $L^{-1}$) | Direct experimental measurement |
| Ni | nitrite concentration (nmol $L^{-1}$) | Direct experimental measurement |
| Na | nitrate concentration (nmol $L^{-1}$) | Direct experimental measurement |
| n15A | nmol $L^{-1}$ of $^{15}NH_4$ | Direct experimental measurement |
| n15Ni | nmol $L^{-1}$ of $^{15}NO_2$ | Direct experimental measurement |
| n15Na | nmol $L^{-1}$ of $^{15}NO_3$ | Direct experimental measurement |
| $R_A$ | Atom % ratio of $^{15}NH_4$ or n15A/A x100 | Direct experimental measurement |
| $R_{Ni}$ | Atom % ratio of $^{15}NO_2$ or n15Ni/Ni x100 | Direct experimental measurement |
| $R_{Na}$ | Atom % ratio of $^{15}NO_3$ or n15Na/Na x100 | Direct experimental measurement |
| $m$ | Remineralization rate ($h^{-1}$ $L^{-1}$) | Estimated with ODE model |
| $u$ | Uptake rate coefficient ($h^{-1}$) | Estimated with ODE model |
| $h$ | Ammonium oxidation rate coefficient ($h^{-1}$) | Estimated with ODE model |
| $x$ | Nitrite oxidation rate coefficient ($h^{-1}$) | Estimated with ODE model |
| $r$ | Nitrite reduction rate coefficient ($h^{-1}$) | Estimated with ODE model |
| $y$ | Nitrate reduction rate coefficient ($h^{-1}$) | Estimated with ODE model |



Table 2. A statistical summary of the role of mussels and day versus night on resulting seawater chemistry and temperature immediately prior to tidepool re-inundation. We used linear mixed effects models with tidepool as a random effect and log-transformed estimates for nutrient concentration; t values are given; §=0.10>p>0.05*=p<0.05, **p<0.001. The number of observations was 40.

| | Mussels | Time of Day | Mussels* Time of Day |
|---|---|---|---|
| $[NH_4^+]$ | 3.076* | 4.225** | 0.841 |
| $[NO_2^-]$ | 2.421* | 0.232 | 2.327* |
| $[NO_3^-]$ | 1.865§ | 0.327 | 1.086 |
| Percent $O_2$ | 2.727* | 6.913** | 2.045§ |
| Temperature | 0.784 | 9.254** | 0.950 |
| pH | 2.223§ | 3.716* | 1.613 |




Table 3. A summary of all estimated rates by treatment in the ODE model (Eqs. 2-7).

Means and (se) are shown with n=10 per treatment. The contribution of microbial

transformations to overall ammonium and nitrate fluxes was quantified as the percentage

that microbial activity ($NH_4^+$ oxidation, $NO_3^-$ reduction, $NO_2^-$ oxidation and reduction)

contributed to all nitrogen uptake, including nitrogen uptake of phototrophs ($u$).

| Rates (nmol L$^{-1}$ hr$^{-1}$) | Mussels | | No Mussels | |
|---|---|---|---|---|
| | day | Night | day | night |
| Ammonium oxidation ($h\overline{A}$) | 11695 (5945) | 490 (262) | 1435 (572) | 161 (145) |
| Nitrite oxidation ($x\overline{Ni}$) | 6980 (2433) | 1904 (1173) | 867 (267) | 148 (140) |
| Nitrate reduction ($y\overline{Na}$) | 4548 (2098) | 2261 (1284) | 435 (197) | 34 (12) |
| Nitrite reduction ($r\,\overline{Ni}$) | 9170 (5281) | 298 (286) | 1228 (649) | 2 (2) |
| Remineralization ($m$) | 25079 (4554) | 7082 (3229) | 6471 (1308) | 3017 (868) |
| Ammonium uptake ($2u\overline{A}$) | 20414 (4103) | 5279 (2676) | 4904 (1131) | 2405 (618) |
| Nitrate uptake ($u\overline{Na}$) | 3206 (530) | 1465 (585) | 1064 (159) | 1140 (324) |
| % ammonium flux due to microbial activity (of total) | 32 (13) | 12 (10) | 22 (9) | 3 (2) |
| % nitrate flux due to microbial activity (of total) | 61 (9) | 30 (18) | 39 (14) | 8 (4) |

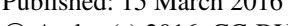


Table 4. A statistical summary of the role of mussels, day versus night, and their

interaction on the rates of nitrogen transformations (in nmol $L^{-1}$ $hr^{-1}$) estimated in both

our ODE models and the traditional source-product models. Linear mixed effects models

using tidepool as a random effect were used on log-transformed or square-root

transformed estimates from Eq. 2-7; t values are given; §=$0.10>p>0.05$, *=$p<0.05$,

**$p<0.001$. The correlation between coefficients estimated from each method is shown

in the last column; no coefficients were significant. There were 40 observations for the

ODE model and 20 for the source-sink model.

| Rate | ODE Model Estimates | | | Source-Product Model | | | |
|---|---|---|---|---|---|---|---|
| | Mussels | Time of Day | Mussels x Time of Day | Mussels | Time of Day | Mussels x Time of Day | Corr |
| **Ammonium oxidation ($h\overline{A}$)** | 3.131* | 4.168** | 2.025* | 2.568* | 1.970§ | 2.080§ | 0.004 |
| **Nitrite oxidation ($x\overline{N\iota}$)** | 2.709* | 5.054** | 2.232* | 1.278 | 0.364 | 0.935 | -0.216 |
| **Nitrate reduction ($y\overline{Na}$)** | 2.725* | 1.205 | 0.774 | 0.761 | 4.103* | 1.657 | -0.021 |
| **Nitrite reduction ($r\,\overline{N\iota}$)** | 2.032§ | 2.907* | 1.209 | 2.561* | 3.365* | 1.172 | -0.010 |
| **Remineralization ($m$)** | 4.139* | 5.676** | 2.722* | | | | |
| **Ammonium uptake ($2u\overline{A}$)** | 4.183* | 5.478** | 2.853* | | | | |
| **Nitrate uptake ($u\overline{\overline{Na}}$)** | 3.336* | 3.323 * | 2.307 * | | | | |

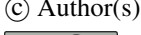

Appendix A1. Example dynamics of stable nitrogen isotopes ($\delta^{15}$N) of tidepool

ammonium, nitrite and nitrate for 4 separate $^{15}$N enrichment experiments made at

different times in a single control tidepool (with mussels). We measured values prior to

the addition of tracer ($T_o$), followed by an immediate post-tracer measurement ($T_1$), and

an approximately 2-3 hour ($T_2$) and a 5-6 hour ($T_3$) post-tracer measurement.  The left 2

panels show the addition of enriched ammonium and the resultant nitrate and nitrite

enrichment, while the right 2 panels show the addition of enriched nitrate and the

resultant enrichment in ammonium and nitrite. In all cases, the $\delta^{15}$N (‰) axis scale for

the enriched source is double that of the product quantities.

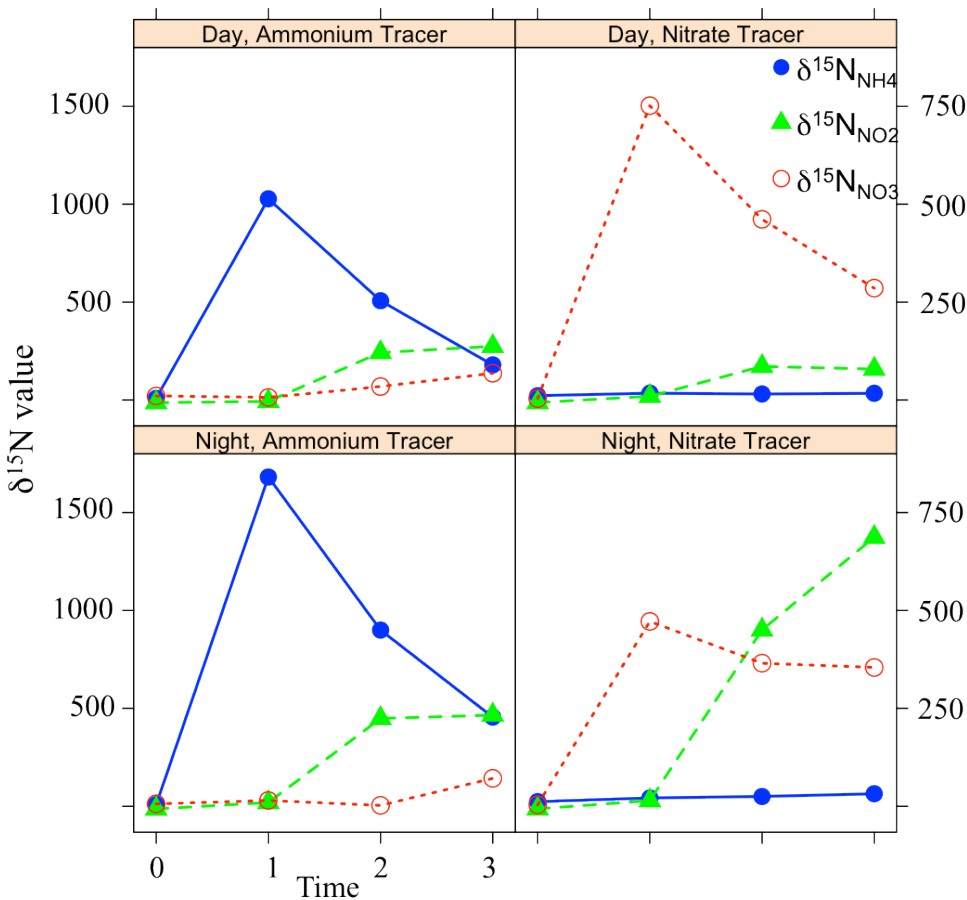