# Peer review of "Tracer experiment and model evidence for macrofaunal shaping of microbial"

_Biogeosciences, 2016_

## Referee Comment (RC1) · Anonymous Referee #1 · 22 Apr 2016

General Comments The authors present a study in which they evaluated the potential relationship between macrofauna and nitrogen transformation by microbes in rocky intertidal systems. The authors used 15N tracer addition experiments in tide pools, which they treated as natural mesocosms, to test the role of mussels and light on microbial nitrogen processing. Rather than using the traditional method of calculating ammonium, nitrate, and nitrite processing via the source-product model, the authors used a series of ordinary differential equations (ODE) to quantify multiple, simultaneous nitrogen transformations. The authors found that mussels enhanced nearly every nitrogen cycling process measured, and that rates were often further enhanced in the daytime. They also found that their nitrogen cycling rates calculated via ODE were

always greater than the rates calculated using prior methods, suggesting that the older method would have significantly underestimated actual rates in this study. Importantly, their results highlight the significant role that isotope dilution can play in contributing to error in these calculations, and the ODE model should be used in future studies.

Overall, I think the authors addressed important questions related to isotope tracer methodology as well as ecology and biogeochemistry that will be of interest to many readers of this journal. The paper is very well written, and the ODEs and related calculations are explained so clearly. I am comfortable with the conclusions and support publication of this manuscript with minor edits, as detailed below. This paper was a pleasure to read.

Specific Comments

lines 180-181 It would be helpful to get an idea of how much the value of the multiplier (2:1 for NH4:NO3 uptake) influences the fits. It's useful that 2:1 fit the data best, but does empirical evidence exist for what this rate is in nature? Please clarify.

line 437 replace "no" with "not"

line 476 insert "which" after "transformations"

Fig. 2 I suggest adding labels to each panel, which you can refer to in the results and discussion more explicitly. You have this in figure 3, and I don't think it's too busy. I also think changing the legend from "mussel control" just to "mussels" would be clearer. These are simply stylistic recommendations, but they would have made things a bit clearer for me.

Fig. 5 See comments for Fig. 2. Also, what is the purpose for the blue shading? I suggest just using grey for consistency.

Table 2 In the caption, you need to add a comma after "p>0.05".

Appendix A1 I suggest using the actual time axis (in hours) rather than the categorical

axis of T1, T2, T3, etc. This will more accurately represent 15N dynamics. You have done this already in Fig. 3.

---

## Referee Comment (RC2) · Anonymous Referee #2 · 25 Apr 2016

Pfister et al. describe an isotope tracer study to unravel N mineralisation and subsequent N cycling in tidal pools of rocky shores, in which they distinguish amongst the activity of mussels (and associated microbes), microbes and phototrophs. To interpret the results, the authors present a new model to account for isotope dilution and continuous N cycling which proves very useful to recover N transformation rates from the available observations. Although the study is well designed, the manuscript would benefit from a clearer description of the components in the system and their interactions that the authors consider. Below I make suggestions to improve the clarity of the manuscript. Overall however this is a very nice study that deserves to be published in Biogeosciences.

Major points:
1. There are many missing references in the literature list, which was quite annoying. Just to name a few: Worm et al. 2006, Layman et al. 2010, Stief 2013 and Heisterkamp et al. 2013. Please check whole paper thoroughly.
2. The Introduction and the first section of the Materials and Methods should be merged. The introduction is quite general, with terms such as nitrogen regeneration, remineralisation, transformation, 'interacting contributions of microbes and macrobiota', competition between phototrophs and fauna-hosted nitrogen-metabolizing microbes and so on. Unfortunately, it doesn't get clear in the introduction what the authors are actually studying and which components of the rocky shore they are targeting. In fact, the clearest part of the 'intro' is given the first section of the Material and Methods section (Lines 111-126), where amongst others Fig. 1 is introduced. Therefore, I suggest to shorten the Introduction and merge the first part of the Method section with the Introduction.
3. Potential overlap with Pather et al. 2014. On lines 218-219, it is indicated that detailed Methods are given in Pather et al. 2014, but it is also important to indicate whether there is data overlap between the present manuscript and Pather et a. 2014. If so, it should be clarified what the novelty (apart from the new model) is of the present study.
4. Units of N transformation rates. The rates are presently expressed in L-1, it seems that transformation rates are actually mediated by the benthos (either mussels or benthic algae uptake [Line 394-395]. Hence, I suggest to express all rates per m2 instead of per L. Firstly to standardise for the depth differences among tidepools and secondly to ease comparison of N cycling rates with other hard- and soft-bottom ecosystems (which I would encourage).

Minor points
1. Line 24: What is meant with "interdependency" here? It would be useful to make a clearer distinction throughout the manuscript among the microbial components considered: fauna-associated microbes and biofilm/sediment-associated microbes (and phototrophs?). Lines 33-35 would already benefit from a clearer distinction.
2. Line 30: The term 'animal presence' is unclear here, because do the authors refer to the activity of fauna (e.g. NH4 release) that influences N cycling or the fact that fauna-associated microbes are present or both?
3. Line 122: The authors first describe that traditional source-sink models are inadequate in the present situation (lines 115-120) and then the first model they describe for their study is such a source-sink model. I suggest to present the new model as (1) and the source-sink model as (2).
4. Line 166: change "multiple, simultaneous processes" to "multiple processes simultaneously".
5. Line 171: The notation of concentration in brackets is awkward. Why not use the common square brackets for concentrations?
6. Line 235: I guess the delta values have been converted to atom% (see Line 149) in stead of the 15N/14N ratio. Note that the [15N] is calculated from atom% x nutrient concentration.
7. Line 265: briefly describe the methods for nutrient and isotope composition here.
8. Line 299: I do not understand the replication in this study. There were 5 tidepools with and 5 tidepools without mussels (Line 220), but I can't find back how the NH4 and NO3 treatments were divided over these tidepools. E.g. were the same pools used for day- and nighttime experimentation, if so, was it checked whether label from earlier experiment was gone?

9. Line 274-277: The authors used the fitting routine modFit to estimate the parameters. Were the field observations weighed in the model cost to account for the differences in units (and absolute values)? I.e. any fitting routine penalises deviations between model and data stronger for variable with high values (e.g. [NH4] is >> [n15Ni], so a model-data deviation for [NH4] weighs heavier on the model cost as compared to a deviation of [n15Ni]).

10. Line 284-286. The convergence to the same sum of squared deviations (SSD) is indeed a good indication that the fitting routine found *a* best parameter set, but it can still be that the same SSD is reached with different sets of best model parameters. Have the authors also checked whether also model parameters converged after the fitting?

11. Line 299-301. Were stats done on the rate parameters (which are concentration independent, e.g. u) or on transformation rates (i.e. 2*u*[NH4]).

12. Line 339-343: The percentage calculations discussed here are unclear, please rephrase and refer to Fig. 1 or the rate parameters.

13. Line 359: So were the 'concentration-independent' model parameters compared or the total nitrogen transformation rates?

14. Line 420: …observation… should be …observations…

15. Line 420-422: unclear sentence (operation of other N processing pathways?), please clarify.

16. Line 453-455: Remineralisation rates dropped strongly during the night. Do the authors have an explanation for this strong reduction? Oxygen and temperature are lower during the night, but is this sufficient to explain the drop?

17. Line 496: Why do the authors suggest that measurement error is the prime factor for the observed variability among tidepools?

18. Fig 1. should have 2*u for ammonium uptake u for nitrate uptake as described in the text.

---

## Author Comment (AC1) · 16 May 2016

To the Editor
BioGeoSciences                                                    16 May 2016

Dear Dr . Middelburg:

We thank the reviewers for their thoughtful comments and provide a reply to each of them. We hope that our revisions further clarify the presentation. As per the instructions, we will not submit a revised manuscript until we hear from the Editor.

Respectfully submitted,
Catherine Pfister & on behalf of co-authors.

*Anonymous Referee #1*

*General Comments The authors present a study in which they evaluated the potential relationship between macrofauna and nitrogen transformation by microbes in rocky intertidal systems. The authors used 15N tracer addition experiments in tide pools, which they treated as natural mesocosms, to test the role of mussels and light on microbial nitrogen processing. Rather than using the traditional method of calculating ammonium, nitrate, and nitrite processing via the source-product model, the authors used a series of ordinary differential equations (ODE) to quantify multiple, simultaneous nitrogen transformations. The authors found that mussels enhanced nearly every nitrogen cycling process measured, and that rates were often further enhanced in the daytime. They also found that their nitrogen cycling rates calculated via ODE were always greater than the rates calculated using prior methods, suggesting that the older method would have significantly underestimated actual rates in this study. Importantly, their results highlight the significant role that isotope dilution can play in contributing to error in these calculations, and the ODE model should be used in future studies.*
*Overall, I think the authors addressed important questions related to isotope tracer methodology as well as ecology and biogeochemistry that will be of interest to many readers of this journal. The paper is very well written, and the ODEs and related calculations are explained so clearly. I am comfortable with the conclusions and support publication of this manuscript with minor edits, as detailed below. This paper was a pleasure to read.*

We thank the reviewer for their time and compliments.

*Specific Comments*
*lines 180-181 It would be helpful to get an idea of how much the value of the multiplier (2:1 for NH4:NO3 uptake) influences the fits. It's useful that 2:1 fit the data best, but does empirical evidence exist for what this rate is in nature? Please clarify.*

There is little guidance in the literature for how we might quantitatively model the relative preference of phototrophs for ammonium versus nitrate in nature. We know that ammonium is energetically easier to take up and there is evidence that tidepool algae readily take up both. The ratio of 2:1 ammonium: nitrate uptake we used broadly reflected this knowledge, but also fit the data well. As an illustration, we present a figure of one tidepool with ammonium enrichment and

show how varying the ratio of $u$ affected the fit of the model to data. This figure is now a new Appendix A2, shown below.

[Figure]

Appendix A2. We show the fit of models (colored lines) to data (points) where we varied the ratio of $u$ for ammonium uptake versus nitrate uptake for a single tidepool during a daytime, ammonium enrichment. This representative scenario shows that a 2:1 ratio, where phototrophic ammonium uptake is twice that of nitrate uptake, was the best fitting model. Greater or lesser ratios did not fit the data as well. Other figure attributes as in Fig 3.

*line 437 replace "no" with "not"*
Done

*line 476 insert "which" after "transformations"*
Done

*Fig. 2 I suggest adding labels to each panel, which you can refer to in the results and discussion more explicitly. You have this in figure 3, and I don't think it's too busy. I also think changing the legend from "mussel control" just to "mussels" would be clearer. These are simply stylistic recommendations, but they would have made things a bit clearer for me.*

Revised as suggested, illustrated below and now included in the revised ms.

[Figure]

*Fig. 5 See comments for Fig. 2. Also, what is the purpose for the blue shading? I suggest just using grey for consistency.*

We reworded to 'mussels'. The blue was to signify that these estimates came from a distinct model (source-sink). For the ODE model, we used the gray color scheme in Fig 5 and Fig 2.

*Table 2 In the caption, you need to add a comma after "p>0.05".*

Done

*Appendix A1 I suggest using the actual time axis (in hours) rather than the categorical axis of T1, T2, T3, etc. This will more accurately represent 15N dynamics. You have done this already in Fig. 3.*

Revised as suggested, and illustrated below.

[Figure]

*Reviewer 2*

*Pfister et al. describe an isotope tracer study to unravel N mineralisation and subsequent N cycling in tidal pools of rocky shores, in which they distinguish amongst the activity of mussels (and associated microbes), microbes and phototrophs. To interpret the results, the authors present a new model to account for isotope dilution and continuous N cycling which proves very useful to recover N transformation rates from the available observations. Although the study is well designed, the manuscript would benefit from a clearer description of the components in the system and their interactions that the authors consider. Below I make suggestions to improve the clarity of the manuscript. Overall however this is a very nice study that deserves to be published in Biogeosciences.*

We thank the reviewer for their close reading of the ms and their compliments. We have revised the introduction, with consideration of the advice below. We note, however, that rev 1 stated "*The paper is very well written, and the ODEs and related calculations are explained so clearly*". We thus tried to keep our edits targeted toward reviewer 2's specific concerns.

*Major points:*
*1. There are many missing references in the literature list, which was quite annoying. Just to name a few: Worm et al. 2006, Layman et al. 2010, Stief 2013 and Heisterkamp et al. 2013. Please check whole paper thoroughly.*

There were many mistakes in this reference list. Apologies and corrections have been made.

*2. The Introduction and the first section of the Materials and Methods should be merged. The introduction is quite general, with terms such as nitrogen regeneration, remineralisation, transformation, 'interacting contributions of microbes and macrobiota', competition between phototrophs and fauna-hosted nitrogen-metabolizing microbes and so on. Unfortunately, it doesn't get clear in the introduction what the authors are actually studying and which components of the rocky shore they are targeting. In fact, the clearest part of the 'intro' is given the first section of the Material and Methods section (Lines 111-126), where amongst others Fig. 1 is introduced. Therefore, I suggest to shorten the Introduction and merge the first part of the Method section with the Introduction.*
As suggested, we have moved the section that introduces the shortcoming of the current modeling approach and the advantages of ours further up into the introduction. We have also made edits to the introduction in an attempt to make it more succinct. These edits are located at L 91-109.

*3. Potential overlap with Pather et al. 2014. On lines 218-219, it is indicated that detailed Methods are given in Pather et al. 2014, but it is also important to indicate whether there is data overlap between the present manuscript and Pather et a. 2014. If so, it should be clarified what the novelty (apart from the new model) is of the present study.*

We have made edits to indicate that Pather et al. 2014 did not explicity quantify microbial transformations and did not report nitrate enrichment studies. L224-227.

*4. Units of N transformation rates. The rates are presently expressed in L-1, it seems that transformation rates are actually mediated by the benthos (either mussels or benthic algae uptake [Line 394-395]. Hence, I suggest to express all rates per m2 instead of per L. Firstly to standardise for the depth differences among tidepools and secondly to ease comparison of N cycling rates with other hard- and soft-bottom ecosystems (which I would encourage).*

The reviewer makes a good point that the area of the benthos (as well as the species composition) likely has a strong effect on the rates of nitrogen transformations. We did not express rates on an areal basis, however, because: 1) All rates in the literature are volumetric and we wanted to compare to these, and 2) Quantifying the area in terms of benthic components would have been logistically difficult. We would have needed both 3D mapping and precise species estimate, and we have neither. Instead, the focus of other research efforts (CAP and MA,

research in progress) is quantifying rates on a per individual and species basis so we can scale up.

*Minor points*
*1. Line 24: What is meant with "interdependency" here? It would be useful to make a clearer distinction throughout the manuscript among the microbial components considered: fauna associated microbes and biofilm/sediment-associated microbes (and phototrophs?). Lines 33-35 would already benefit from a clearer distinction.*
Reworded, L34

*2. Line 30: The term 'animal presence' is unclear here, because do the authors refer to the activity of fauna (e.g. NH4 release) that influences N cycling or the fact that fauna-associated microbes are present or both?*
It refers to both, L30

*3. Line 122: The authors first describe that traditional source-sink models are inadequate in the present situation (lines 115-120) and then the first model they describe for their study is such a source-sink model. I suggest to present the new model as (1) and the source-sink model as (2).*
Although we understand the logic here, it was difficult to present the more complicated 6 equation model without first showing the simpler, single equation model. In other words, presenting how our model was unique was difficult without first presenting what model was being utilized. We thus left the presentation as it was and hope it is clear to the readers. L143

*4. Line 166: change "multiple, simultaneous processes" to "multiple processes simultaneously".*
Done.

*5. Line 171: The notation of concentration in brackets is awkward. Why not use the common square brackets for concentrations?*
Revised to square brackets.

*6. Line 235: I guess the delta values have been converted to atom% (see Line 149) instead of the 15N/14N ratio. Note that the [15N] is calculated from atom% x nutrient concentration.*
Yes.

*7. Line 265: briefly describe the methods for nutrient and isotope composition here.*
Done. L266-271

*8. Line 299: I do not understand the replication in this study. There were 5 tidepools with and 5 tidepools without mussels (Line 220), but I can't find back how the NH4 and NO3 treatments were divided over these tidepools. E.g. were the same pools used for day- and nighttime experimentation, if so, was it checked whether label from earlier experiment was gone?*
We added some additional text to clarify that we separated the experiments by 2 days, and always estimated enrichment and rates based on $T_o$ samples. L240-242.

*9. Line 274-277: The authors used the fitting routine modFit to estimate the parameters. Were*

*the field observations weighed in the model cost to account for the differences in units (and absolute values)? I.e. any fitting routine penalises deviations between model and data stronger for variable with high values (e.g. [NH4] is >> [n15Ni], so a model-data deviation for [NH4] weighs heavier on the model cost as compared to a deviation of [n15Ni]).*
We assume that the reviewer is asking whether we weighted our parameter estimates based on knowledge of variation. We did not. A good weighting scheme would need to account not only for variance, but also for measurement accuracy. We do not feel that we know these factors enough to alter our methodology.

*10. Line 284-286. The convergence to the same sum of squared deviations (SSD) is indeed a good indication that the fitting routine found a best parameter set, but it can still be that the same SSD is reached with different sets of best model parameters. Have the authors also checked whether also model parameters converged after the fitting?*
Yes, we did check this. Because we did 100 random restarts across initial parameter values, we were able to see whether the model found different parameter sets with the same SSD, and this was not a typical outcome of solving the equations.

*11. Line 299-301. Were stats done on the rate parameters (which are concentration independent, e.g. u) or on transformation rates (i.e. 2\*u\*[NH4]).*
We performed statistics on the transformation rates.

*12. Line 339-343: The percentage calculations discussed here are unclear, please rephrase and refer to Fig. 1 or the rate parameters.*
We now insert text to make it clear how our percentage flux calculations relate to the parameters in Fig 1. L351.

*13. Line 359: So were the 'concentration-independent' model parameters compared or the total nitrogen transformation rates?*
We compared the total nitrogen transformation rates, thus resulting in a measure that is nmol N $L^{-1}$ $h^{-1}$.

*14. Line 420: ...observation... should be ...observations...*
reworded.
*15. Line 420-422: unclear sentence (operation of other N processing pathways?), please clarify.*
reworded.
*16. Line 453-455: Remineralisation rates dropped strongly during the night. Do the authors have an explanation for this strong reduction? Oxygen and temperature are lower during the night, but is this sufficient to explain the drop?*
The metabolic activity of the animals may slow and thus reduce remineralization. Both lower oxygen and lower temperature could lead to reduced animal metabolism and decreased ammonium production. Though we do not know this with certainty, we now suggest the possibility. L467-468.

*17. Line 496: Why do the authors suggest that measurement error is the prime factor for the observed variability among tidepools?*
We did not mean to imply that measurement error has primacy. We mentioned a couple of

factors, including "natural variability in space and time for processes sensitive to species composition and environmental factors." We inserted the word 'both' on L 510 to help make this clear.

*18. Fig 1. should have 2\*u for ammonium uptake u for nitrate uptake as described in the text.*
Changed.

---

## Author Comment (AC2) · 16 May 2016

[revised manuscript text omitted]

Allgeier, J. E., Layman, C. A., Mumby, P. J. and Rosemond, A. D.: Consistent nutrient storage and supply mediated by diverse fish communities in coral reef ecosystems, Global Change

Biology, 20(8), 2459–2472, doi:10.1111/gcb.12566, 2014.

Aquilino, K. M., Bracken, M. E. S., Faubel, M. N. and Stachowicz, J. J.: Local-scale nutrient regeneration facilitates seaweed growth on wave-exposed rocky shores in an upwelling system, Limnology and Oceanography, 54(1), 309–317, doi:10.4319/lo.2009.54.1.0309, 2009.

Atkinson, C. L. and Vaughn, C. C.: Biogeochemical hotspots: temporal and spatial scaling of the impact of freshwater mussels on ecosystem function, Freshw Biol, 60(3), 563–574, doi:10.1111/fwb.12498, 2015.

Bayne, B. L. and Scullard, C.: Rates of nitrogen excretion by species of *Mytilus* (Bivalvia:

Mollusca), Journal of the Marine Biological Association of the United Kingdom, 57(2), 355–

369, doi:10.1017/S0025315400021809, 1977.

Beman, J. Michael, Chow, Cheryl-Emiliane, King, Andrew, Feng, Yuanyuan, Fuhrman, Jed A.,

Andersson, Andreas and Bates, Nicholas R.: Global declines in ocean nitrification rates as a consequence of ocean acidification., Proceedings of the National Academy of Sciences, 108,

208–213, 2011.

Bracken, M. E. S.: Invertebrate-mediated nutrient loading increases growth of an intertidal macroalga, Journal of Phycology, 40(6), 1032–1041, doi:10.1111/j.1529-8817.2004.03106.x,

2004.

Casciotti, K. L.: Inverse kinetic isotope fractionation during bacterial nitrite oxidation,

Geochimica et Cosmochimica Acta, 73(7), 2061–2076, doi:10.1016/j.gca.2008.12.022, 2009.

Dortch, Q.: The interaction between ammonium and nitrate uptake in phytoplankton, , 61, 183–

201, 1990.

Dugdale, R. and Goering, J.: Uptake of new and regenerated forms of nitrogen in primary productivity, Limnology and Oceanography, 12(2), 196–206, 1967.

Fiore, C. L., Jarett, J. K., Olson, N. D. and Lesser, M. P.: Nitrogen fixation and nitrogen transformations in marine symbioses, Trends in Microbiology, 18(10), 455–463, doi:10.1016/j.tim.2010.07.001, 2010.

Fowler, D., Coyle, M., Skiba, U., Sutton, M. A., Cape, J. N., Reis, S., Sheppard, L. J., Jenkins,

A., Grizzetti, B., Galloway, J. N., Vitousek, P., Leach, A., Bouwman, A. F., Butterbach-Bahl,

K., Dentener, F., Stevenson, D., Amann, M. and Voss, M.: The global nitrogen cycle in the twenty-first century, Philosophical Transactions of the Royal Society of London B: Biological

Sciences, 368(1621), 20130164, doi:10.1098/rstb.2013.0164, 2013.

Galloway, J. N., Townsend, A. R., Erisman, J. W., Bekunda, M., Cai, Z., Freney, J. R.,

Martinelli, L. A., Seitzinger, S. P. and Sutton, M. A.: Transformation of the nitrogen cycle:

recent trends, questions, and potential solutions, Science, 320(5878), 889–892, doi:10.1126/science.1136674, 2008.

Glibert, Pamela M., Lipschultz, F., McCarthy, James J. and Altabet, M. A.: Isotope dilution models of uptake and remineralization of ammonium by marine plankton, Limnology and

Oceanography, 27, 639–650, 1982.

Granger, J., Sigman, D. M., Lehmann, M. F. and Tortell, P. D.: Nitrogen and oxygen isotope fractionation during dissimilatory nitrate reduction by denitrifying bacteria, Limnol.

Oceanogr., 53(6), 2533–2545, doi:10.4319/lo.2008.53.6.2533, 2008.

Granger, J., Sigman, D. M., Rohde, M. M., Maldonado, M. T. and Tortell, P. D.: N and O

isotope effects during nitrate assimilation by unicellular prokaryotic and eukaryotic plankton cultures, Geochimica et Cosmochimica Acta, 74(3), 1030–1040, doi:10.1016/j.gca.2009.10.044, 2010.

Guerrero, M. A. and Jones, R. D.: Photoinhibition of marine nitrifying bacteria: wavelength-dependent response., Mar Ecol Prog Ser, 141, 183–192, 1995.

Heisterkamp, I. M., Schramm, A., Larsen, L. H., Svenningsen, N. B., Lavik, G., de Beer, D. and Stief, P.: Shell biofilm-associated nitrous oxide production in marine molluscs: processes, precursors and relative importance: Nitrous oxide production in shell biofilms, Environmental Microbiology, 15(7), 1943–1955, doi:10.1111/j.1462-2920.2012.02823.x, 2013.

Horrigan, S. G. and Springer, A. L.: Oceanic and estuarine ammonium oxidation: Effects of light, Limnology and Oceanography, 35(2), 479–482, doi:10.4319/lo.1990.35.2.0479, 1990.

Hurd, C. L., Harrison, P. J., Bischof, K. and Lobban, C. S.: Seaweed ecology and physiology, Second edition., Cambridge University Press, Cambridge ; New York., 2014.

Kremer, B. P.: Metabolic implications of non-photosynthetic carbon fixation in brown macroalgae, Phycologia, 20(3), 242–250, doi:10.2216/i0031-8884-20-3-242.1, 1981.

Lipschultz, F.: Isotope tracer methods for studies of the marine nitrogen cycle, in Nitrogen in the Marine Environment (2nd Edition), edited by D. G. Capone, D. A. Bronk, M. R. Mulholland, and E. J. Carpenter, pp. 1345–1384, Academic Press, San Diego. [online] Available from: http://www.sciencedirect.com/science/article/pii/B9780123725226000311 (Accessed 19 May 2014), 2008.

Magalhaes, C. M., Bordalo, A. A. and Wiebe, W. J.: Intertidal biofilms on rocky substratum can play a major role in estuarine carbon and nutrient dynamics, Mar Ecol Prog Ser, 258, 275–281, doi:10.3354/meps258275, 2003.

Maranger, R., Caraco, N., Duhamel, J. and Amyot, M.: Nitrogen transfer from sea to land via commercial fisheries, Nature Geosci, 1(2), 111–112, doi:10.1038/ngeo108, 2008.

McIlvin, M. R. and Altabet, M. A.: Chemical conversion of nitrate and nitrite to nitrous oxide for nitrogen and oxygen isotopic analysis in freshwater and aeawater, Analytical Chemistry,

77(17), 5589–5595, doi:10.1021/ac050528s, 2005.

Miranda, L. N., Hutchison, K., Grossman, A. R. and Brawley, S. H.: Diversity and abundance of the bacterial community of the red macroalga *Porphyra umbilicalis*: did bacterial farmers produce macroalgae?, edited by J. Neufeld, PLoS ONE, 8(3), e58269, doi:10.1371/journal.pone.0058269, 2013.

Moulton, O. M., Altabet, M. A., Beman, J. M., Deegan, L. A., Lloret, J., Lyons, M. K., Nelson,

J. A. and Pfister, C. A.: Microbial associations with macrobiota in coastal ecosystems:

patterns and implications for nitrogen cycling, Frontiers in Ecology and the Environment,

14(4), 200–208, doi:10.1002/fee.1262, 2016.

Pather, S., Pfister, C. A., Post, D. M. and Altabet, M. A.: Ammonium cycling in the rocky intertidal: Remineralization, removal, and retention, Limnology and Oceanography, 59(2),

361–372, doi:10.4319/lo.2014.59.2.0361, 2014.

Pfister, C. A.: Estimating competition coefficients from census data: a test with field manipulations of tidepool fishes, The American Naturalist, 146(2), 271–291, 1995.

Pfister, C. A.: Intertidal invertebrates locally enhance primary production, Ecology, 88(7), 1647–

1653, doi:10.1890/06-1913.1, 2007.

Pfister, C. A., Meyer, F. and Antonopoulos, D. A.: Metagenomic profiling of a microbial assemblage associated with the California mussel: A node in networks of carbon and nitrogen cycling, edited by R. DeSalle, PLoS ONE, 5(5), e10518, doi:10.1371/journal.pone.0010518, 2010.

Pfister, C. A., Altabet, M. A. and Post, D.: Animal regeneration and microbial retention of nitrogen along coastal rocky shores, Ecology, 95(10), 2803–2814, doi:10.1890/13-1825.1, 2014a.

Pfister, C. A., Gilbert, J. A. and Gibbons, S. M.: The role of macrobiota in structuring microbial communities along rocky shores, PeerJ, 2, e631, doi:10.7717/peerj.631, 2014b.

Plaganyi, E. E. and Branch, G. M.: Does the limpet Patella cochlear fertilize its own algal garden?, Marine Ecology Progress series, 194, 113–122, 2000.

Risgaard-Petersen, N., Nicolaisen, M. H., Revsbech, N. P. and Lomstein, B. A.: Competition between ammonia-oxidizing bacteria and benthic microalgae, Appl. Environ. Microbiol., 70(9), 5528–5537, doi:10.1128/AEM.70.9.5528-5537.2004, 2004.

Schindler, D. E., Knapp, R. A. and Leavitt, P. R.: Alteration of nutrient cycles and algal production resulting from fish introductions into mountain lakes, Ecosystems, 4(4), 308–321, doi:10.1007/s10021-001-0013-4, 2001.

Soetaert, K., Petzoldt, T. and Dresden, T. U.: Inverse modelling, sensitivity and Monte Carlo Analysis in R using package FME, Journal of Statistical Software, 33(3), 2010.

Soetaert, K., Cash, J. and Mazzia, F.: Solving Differential Equations in R, Springer Berlin Heidelberg, Berlin, Heidelberg. [online] Available from: http://link.springer.com/10.1007/978-3-642-28070-2 (Accessed 14 May 2015), 2012.

Stief, P.: Stimulation of microbial nitrogen cycling in aquatic ecosystems by benthic macrofauna: mechanisms and environmental implications, Biogeosciences, 10(12), 7829–7846, doi:10.5194/bg-10-7829-2013, 2013.

Stief, P., Poulsen, M., Nielsen, L. P., Brix, H. and Schramm, A.: Nitrous oxide emission by

    aquatic macrofauna, Proceedings of the National Academy of Sciences, 106(11), 4296–4300,

    doi:10.1073/pnas.0808228106, 2009.

Subalusky, A. L., Dutton, C. L., Rosi-Marshall, E. J. and Post, D. M.: The hippopotamus

    conveyor belt: vectors of carbon and nutrients from terrestrial grasslands to aquatic systems in

    sub-Saharan Africa, Freshw Biol, 60(3), 512–525, doi:10.1111/fwb.12474, 2015.

Suchanek, Thomas H.: The *Mytilus californianus* community: studies on the composition,

    structure, organization and dynamics of a mussel bed., University of Washington, Seattle,

    PhD Thesis., 1979.

Swart, P. K., Evans, S., Capo, T. and Altabet, M. A.: The fractionation of nitrogen and oxygen

    isotopes in macroalgae during the assimilation of nitrate, Biogeosciences, 11(21), 6147–6157,

    doi:10.5194/bg-11-6147-2014, 2014.

Taylor, R. B. and Rees, T. A. V.: Excretory products of mobile epifauna as a nitrogen source for

    seaweeds, Limnology and Oceanography, 43(4), 600–606, doi:10.4319/lo.1998.43.4.0600,

    1998.

Thomas, T. E. and Harrison, P. J.: Effect of nitrogen supply on nitrogen uptake, accumulation

    and assimilation in *Porphyra perforata* (Rhodophyta), Marine Biology, 85(3), 269–278,

    doi:10.1007/BF00393247, 1985.

Vanni, M. J.: Nutrient cycling by animals in freshwater ecosystems, Annual Review of Ecology

    and Systematics, 33(1), 341–370, doi:10.1146/annurev.ecolsys.33.010802.150519, 2002.

Ward, B. B.: Nitrification, in Nitrogen in the Marine Environment, edited by D. G. Capone, D.

    A. Bronk, M. R. Mulholland, and E. J. Carpenter, pp. 199–262, Elsevier, Amsterdam., 2008.

Welsh, D. T. and Castadelli, G.: Bacterial nitrification activity directly associated with isolated

   benthic marine animals, Marine Biology, 144(5), 1029–1037, doi:10.1007/s00227-003-1252-

   z, 2004.

Worm, B., Barbier, E. B., Beaumont, N., Duffy, J. E., Folke, C., Halpern, B. S., Jackson, J. B. C.,

   Lotze, H. K., Micheli, F., Palumbi, S. R., Sala, E., Selkoe, K. A., Stachowicz, J. J. and

   Watson, R.: Impacts of Biodiversity Loss on Ocean Ecosystem Services, Science, 314(5800),

   787–790, doi:10.1126/science.1132294, 2006.

[revised manuscript text omitted]

Appendix A2. We show the fit of models (colored lines) to data (points) where we varied the ratio of $u$ for ammonium uptake versus nitrate uptake for a single tidepool during a daytime, ammonium enrichment. This representative scenario shows that a 2:1 ratio, where phototrophic ammonium uptake is twice that of nitrate uptake, was the best fitting model. Greater or lesser ratios did not fit the data as well. Other figure attributes as in Fig 3.